# An in-situ daily dataset for benchmarking temporal variability of groundwater recharge

Pragnaditya Malakar[1,2], Aatish Anshuman[1], Mukesh Kumar[1], Georgios Boumis[1], T. Prabhakar Clement[1], Arik Tashie[1], Hitesh Thakur[1], Nagaraj Bhat[1,3], Lokendra Rathore[1]

[1]Department of Civil, Construction, and Environmental Engineering, University of Alabama, Tuscaloosa, AL, USA

[2]Department of Geological Sciences, Jadavpur University, Kolkata, India

[3]Shri Madhwa Vadiraja Institute of Technology and Management, India

*Correspondence to:* Mukesh Kumar (mkumar4@eng.ua.edu) and Pragnaditya Malakar (pragnadityamalakar@gmail.com)

**Abstract.** Accurate estimate of groundwater recharge is crucial for prediction of groundwater table dynamics and dependent eco-hydrological processes. Despite its importance, benchmark data for groundwater recharge at fine (~ daily) temporal resolution is lacking. We present a first-of-a-kind daily groundwater recharge per unit specific yield (RpSy) data over periods of 2 – 38 years at 485 groundwater monitoring wells in the US. The RpSy data for these locations are calculated from the daily groundwater table time series using the water table fluctuations (WTF) method. Although direct validation of the data is not possible, since it is the first of its kind, we compare the RpSy data with the monthly USGS product to identify similarities and differences. The RpSy dataset may serve as a benchmark for validating the temporal consistency of recharge products and daily simulation results from land surface and integrated hydrologic models. The dataset is available at https://doi.org/10.5281/zenodo.13323242 (Malakar et al., 2024).

## 1 Introduction

Groundwater is the largest distributed inland freshwater storage. It sustains human water consumption and acts as a buffer during droughts (Famiglietti, 2014; Seo et al., 2018; Taylor et al., 2013b). In the US, it is estimated that groundwater supplies approximately 60% of irrigation and 40% of the public water (Russo and Lall, 2017). Furthermore, more than 40 million people in the US, a majority of them in rural areas, rely on domestic boreholes for their drinking water demands (Alley et al., 1999; Wu et al., 2021). Alarmingly, mismanagement of freshwater resources, climate variability and change, rapid population growth, and changes in land use due to urbanization and agricultural expansion has put significant stress on groundwater resources. For example, recent studies have shown that the cumulative volume of groundwater in the US has decreased by almost 1,000 km$^3$ between 1900 and 2008 (Konikow, 2015). In certain locations, such as the Mississippi embayment section of the Gulf Coastal Plain, groundwater depletion has occurred at a high rate of 1.2 km$^3$/yr (Konikow, 2015).

To sustainably manage groundwater resources and for conducting more accurate aquifer vulnerability assessment, it is of critical importance that stakeholders and policy managers are informed of the present and future trends of groundwater recharge (GWR), a flux describing the volume of water infiltrating into an aquifer, per unit area, per unit time, [L/T]). Despite the importance of GWR in controlling groundwater level (GWL) dynamics, its accurate estimation remains a challenge. This is in part because GWR is influenced by a range of variables, such as climate forcings, topography, hydrogeology, land cover, land-use patterns, pumping, and antecedent groundwater/soil states (Berghuijs et al., 2022; Chen et al., 2020; Kumar and Duffy, 2015). Furthermore, GWR varies temporally at multiple temporal scales, ranging from event to inter-annual durations (Crosbie et al., 2005; Scanlon et al., 2006; Varni et al., 2013; Squeo et al., 2006; Valois et al., 2020). Depending on the dominant processes and their trends, GWR may show site-specific variations. For example, studies in both humid (Tashie et

al., 2016) and arid (Thomas et al., 2016) settings showed that at the event scale, the intensity of extreme precipitation events changes the fraction of precipitation that contributes to recharge. Others showed that groundwater resources in East Africa are sustained by anomalously intense seasonal and monthly rainfall (Taylor et al., 2013a). Intense episodic precipitation has been observed to produce a large GWR in sub-Saharan Africa (Cuthbert et al., 2019). Analogous efforts from India indicate that in the southern part of the country, GWR is driven by high-intensity precipitation (Asoka et al., 2018), while higher recharge in the northern fertile Indus-Ganges-Brahmaputra aquifer is due to higher precipitation and pumping-induced GWR (Bhanja et al., 2019; Lapworth et al., 2015; Malakar et al., 2021a, b, c). Past evidence from the Northern High Plains in the US indicates that antecedent soil moisture conditions indeed play a vital role in determining GWR in conjunction with extreme rainfall (Zhang et al., 2016). Given that antecedent soil moisture is also strongly affected by variations in meteorological forcing from daily to longer time scales (Ghannam et al., 2016), so can be expected of event recharge as well. Recharge is also likely to vary with seasonal, inter-annual, and decadal variations in rainfall, temperature, and vapor pressure deficit, which influence evapotranspiration (Liu et al., 2017, 2020; Milly and Dunne, 2016; Mueller and Seneviratne, 2014) and its partitioning (Raghav et al., 2022). An increase in temperature increases evaporative demand and decreases soil moisture (Raghav and Kumar, 2021), which could lead to a reduction in recharge (and aquifer replenishment) (Condon et al., 2020), consequently resulting in groundwater depletion (Singh and Borrok, 2019). In summary, GWR is an integrated response of a range of forcings, processes, and local physiographic properties which are crucial for groundwater sustainability.

There has been a general call to action for improving recharge and baseflow processes in large-scale hydro and Land Surface Models (LSMs) for some time (e.g., (Fan et al., 2019)). The need to improve recharge estimates is further underscored by recent studies such as that by Berghuijs et al. (2022), wherein, based on the global recharge dataset by Moeck et al. (2020), it was reported that LSMs tend to systematically underestimate recharge by a factor of about 2. Gnann et al. (2023) also demonstrated that theoretical and empirically based functional relationships between drivers of recharge and the recharge flux differ significantly across global water models. This underscores the alarming potential of divergent responses from models under future projected changes in drivers. An accurate dataset for GWR can not only assist with the assessment of changes in local hydrologic responses but can also help manage emerging threats to food and water security. Furthermore, it can help benchmark models used for estimating GWR.

Despite its importance, what is currently lacking in the scientific record is a continuous, ground-truthing dataset of GWR at monthly or as fine as a daily resolution. Past efforts have mostly focused on obtaining coarser time-resolution groundwater recharge. For example, base flow discharge (Hung Vu and Merkel, 2019; Meyboom, 1961) and isotopic or chemical tracers (Lapworth et al., 2015; McMahon et al., 2011; Scanlon et al., 2010) have been used to estimate recharge over months to decadal scales. These data have formed the basis for several studies that focused on mapping recharge rates at regional to global scales (Moeck et al., 2020). However, these data provide limited to no information related to event-scale or daily resolution variations in recharge (Tashie et al., 2016). While seepage meters (Scanlon et al., 2002), lysimeters (Gong et al., 2021; Xu and Chen, 2005), and heat tracers (Blasch et al., 2007; Healy, 2010) are capable of calculating recharge over a fine temporal resolution, they are expensive and complicated leading to difficulty in the development of continuous datasets at a large number of locations (Tashie et al., 2016). A useful approach for estimating GWR at a fine temporal resolution is based on the water budget method. However, this approach often does not account for the subsurface storage change and is impacted by uncertainties associated with water budget components such as ET and/or quick flow (Reitz et al., 2017a; Reitz and Sanford, 2019a). Furthermore, basin-scaled empirical relationships modelled for recharge estimation oftentimes do not capture finer spatial resolution variations (Gonzalez et al., 2023). An alternative promising option is to obtain GWR estimates using the water table fluctuation (WTF) method (Bhanja et al., 2019; Healy and Cook, 2002; Nimmo et al., 2015).

The WTF method based on the master recession curve (MRC) yields GWR at a fine temporal resolution while only needing available well hydrograph data, meteorological data, and specific yield (Boumis et al., 2022). Studies have highlighted

its applicability for benchmarking recharge estimates from models (Crosbie et al., 2015), and for further understanding recharge variations and its implications on water resources in diverse hydroclimatic regions (Delin et al., 2007; Nimmo et al., 2015; Nimmo and Perkins, 2018; Tashie et al., 2016). One major challenge with the WTF method for estimating recharge is its reliance on specific yield data, which is not commonly accessible and involves a high level of uncertainty. In a study by Boumis et al. (2022), it was suggested that groundwater recharge may have been overestimated due to specific yield values obtained at locations where the water table is near the land surface. In a study by Crosbie et al. (2019), it was noted that improving estimates of specific yield is crucial for addressing the overestimation of recharge. To circumvent the challenge posed by uncertain specific yield on recharge estimation from WTF method, here we instead generate a benchmark dataset that provides recharge per unit specific yield, also referred to as RpSy henceforth. We generate this dataset after performing detailed quality control to identify well locations that are suited for yielding high-fidelity long-term RpSy estimates. Finally, we compare the estimated RpSy with published monthly recharge estimates provided in other published studies (Reitz and Sanford, 2019a) and assess RpSy variations vis-à-vis its primary hydrometeorological influencers.

## 2 Methods

### 2.1 RpSy estimation using the WTF method

GWL fluctuations recorded at monitoring wells are used to estimate recharge based on the water table fluctuation (WTF) method (Healy and Cook, 2002; Healy, 2010). The general principle of the method is to use the water table rise and fall dynamics to compute recharge in a continuous fashion (Heppner and Nimmo, 2005). Quantitatively, GWR within a specific time interval (daily time scale is used in this study) is evaluated in m/d unit as the product of depth to water level rise occurring in a day ($\Delta H$) and specific yield (Sy) (Figure 1a):

$$GWR = \Delta H \times Sy, \tag{1}$$

Here, $\Delta H$ is the water table rise relative to the predicted water level in the absence of recharge. The water level in the absence of recharge is predicted based on the master recession curve (MRC) (Nimmo et al., 2015; Nimmo and Perkins, 2018), which captures the characteristic recession pattern, i.e., the relation between water table elevation (H) and its rate of decline (dH/dt) (Figure 1b). It is to be noted that the WTF-based groundwater recharge incorporating MRC is not merely the difference in water table height between two time points. For instance, after a recharge event, the groundwater table may rise and then recede. Calculating recharge simply by taking the difference in water table height between the start of the recharge event and a point further along the recession period could yield a small or even negative recharge estimate. In our approach to calculating recharge using the WTF method, we determine the recharge by evaluating the difference in groundwater table height between the current and next time step, adjusting for what the height at the current time step would be if it were receding according to the rate defined by the MRC. This variable, $\Delta H$, is evaluated between two consecutive time steps, 1 day in our case. Notably, the time step is much shorter than the duration of a recharge event. Accurate MRC calculation requires careful identification of the slope element – a short enough interval that can be considered as pure recession. Because of the uncertainty inherent in defining a pure recession interval, we consider a range of slope element lengths equal to 4, 6, 8, and 10 days (Boumis et al., 2022). Similarly, multiple minimum interval durations, viz. 2, 4, 6, and 8 days, between precipitation and the start of recession are also considered. Daily precipitation amount smaller than 0.5 mm precipitation is considered negligible while deriving the MRC (Nimmo and Perkins, 2018). For any given site, the slope element and the minimum interval duration length that provides the maximum MRC correlation is used for ensuing analysis.

Although Sy is a primary constituent in GWR estimation using the WTF method (see Eqn. 1), accurate estimates of Sy are unavailable across the US (Li and Rodell, 2015). The feasibility of obtaining Sy for a large number of locations, given

the large spatial heterogeneity in its magnitude, is impractical. Hence, here we evaluate RpSy, instead of GWR, as a benchmarking product. RpSy is calculated as the ratio GWR/Sy using Eqn. 1.

## 2.2 Data Sources and well selection criteria

Groundwater level (GWL) records for the last four decades (1983-2022) are retrieved from the US Geological Survey's (USGS) National Water Information System web interface (link: https://waterdata.usgs.gov/nwis/gw). The data has a temporal resolution of 1 day and are based on observations at the monitoring wells. The following primary criteria are used to narrow down the stations:

a) Wells that have never and/or are not prone to going dry (Cunningham et al., 2007).

b) Wells with at least two years of continuous daily GWL observations with limited data gaps.

c) Groundwater wells that primarily reflect event-scale GWL changes in response to meteorological forcings. To select wells that likely experience event GWR response to precipitation signals, i.e., where the WTF method is valid, we only select the top 75 percentile wells based on the maximum lag correlation coefficient between RpSy and precipitation time series. Selection of 75th percentile as a threshold is subjective and is obtained after trial and error. The goal is to use a threshold that ensures that wells in the lower percentile of maximum lag correlation coefficient correspond to GWR response that is not event based. Example of a selected and a rejected observation well based on this selection criteria is illustrated in Figure S1.

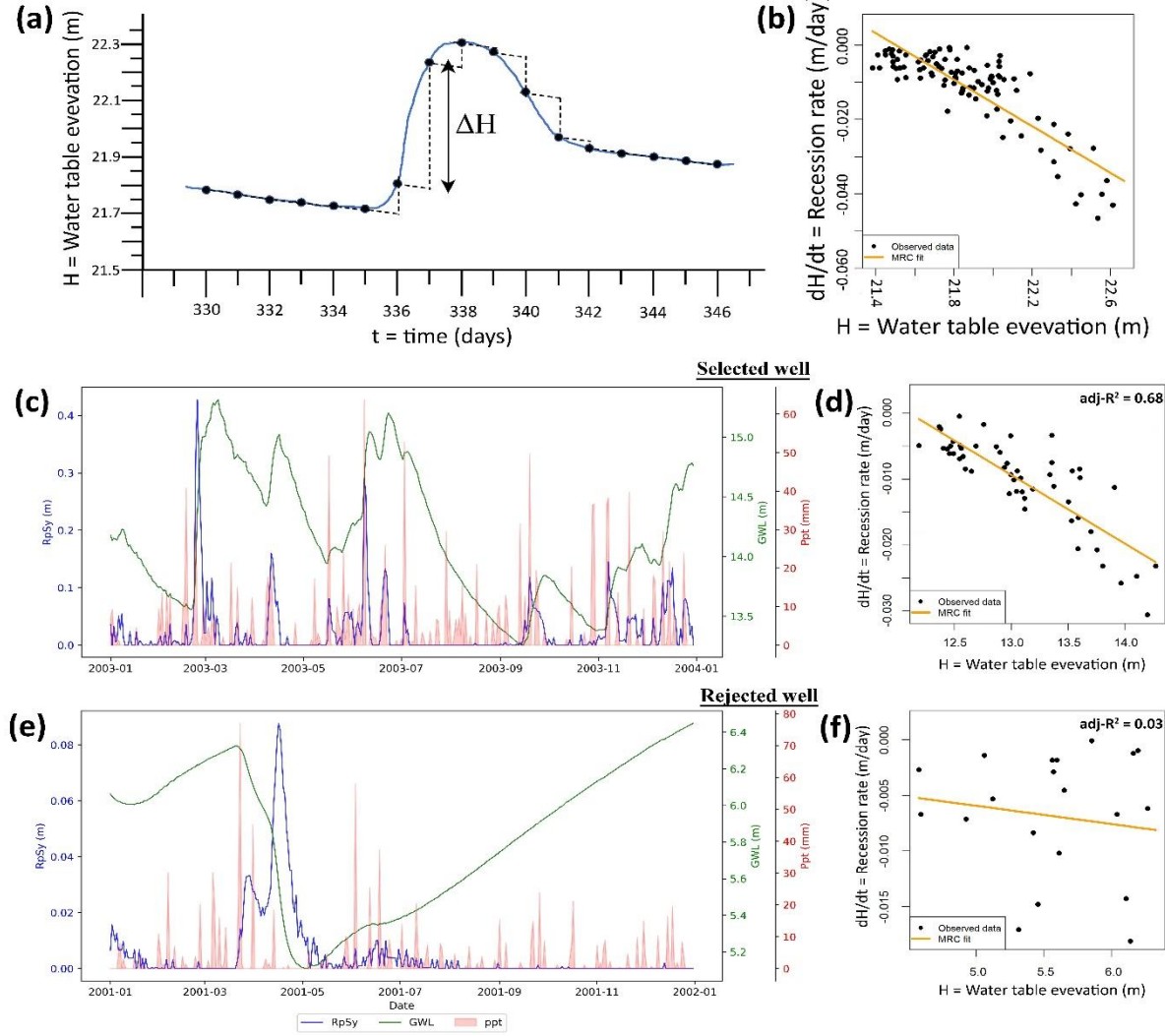

**Figure 1: (a) Schematic representation of ΔH evaluation in the WTF method, which is then used to obtain daily groundwater recharge using Eqn 1, (b) a typical dH/dt vs. H plot used to derive the MRC. The hydrographs (c) and (e) show daily groundwater level (GWL, in m), precipitation (Ppt, in mm), and estimated recharge per unit specific yield (RpSy, in m, discussed later) for a selected and a rejected well, respectively. Here, selection/rejection is based on representativeness of the MRC, which here is determined based on an adj-R2 value of 0.2. (d) and (f) show the dH/dt vs. H plot for the corresponding selected and rejected wells, respectively.**

Next, we further screened the data selected based on the above criteria to develop a subset of wells that met the following secondary criteria including:

i) Well datasets with a characteristic functional relation between H and dH/dt are needed to develop an MRC. To ensure this, an adj-R2 value of 0.2 for the MRC curve fit is used as a threshold. Only the wells with a correlation coefficient above this threshold are considered. It's important to note that sites with high adj-R² values not only confirm the applicability of the WTF method but also help filter out sites with pumping impacts, which can cause variable recession rates. However, this threshold is not foolproof and cannot completely guarantee the absence of groundwater pumping effects.

ii) Finally, since MRC is expected to exhibit a negative slope, only such wells are used in the proceeding analyses.

Examples of selected and rejected observation wells based on the secondary selection criteria are illustrated in Figure 1. After implementation of all these filtering and post-processing criteria, 485 daily observation wells out of 2136 are finally found suitable for our analysis (Figure 2). Most selected wells fall within the eastern part of the US. Precipitation data from PRISM (Daly et al., 2008) are used to obtain the estimates of the MRC characteristics curve. The PRISM data has a temporal resolution of 1 day and a spatial resolution of 4 km. Evapotranspiration data that has been used in the technical validation (discussed later) is obtained from the Global Land Evaporation Amsterdam Model (GLEAM) (Martens et al., 2017; Miralles et al., 2011).

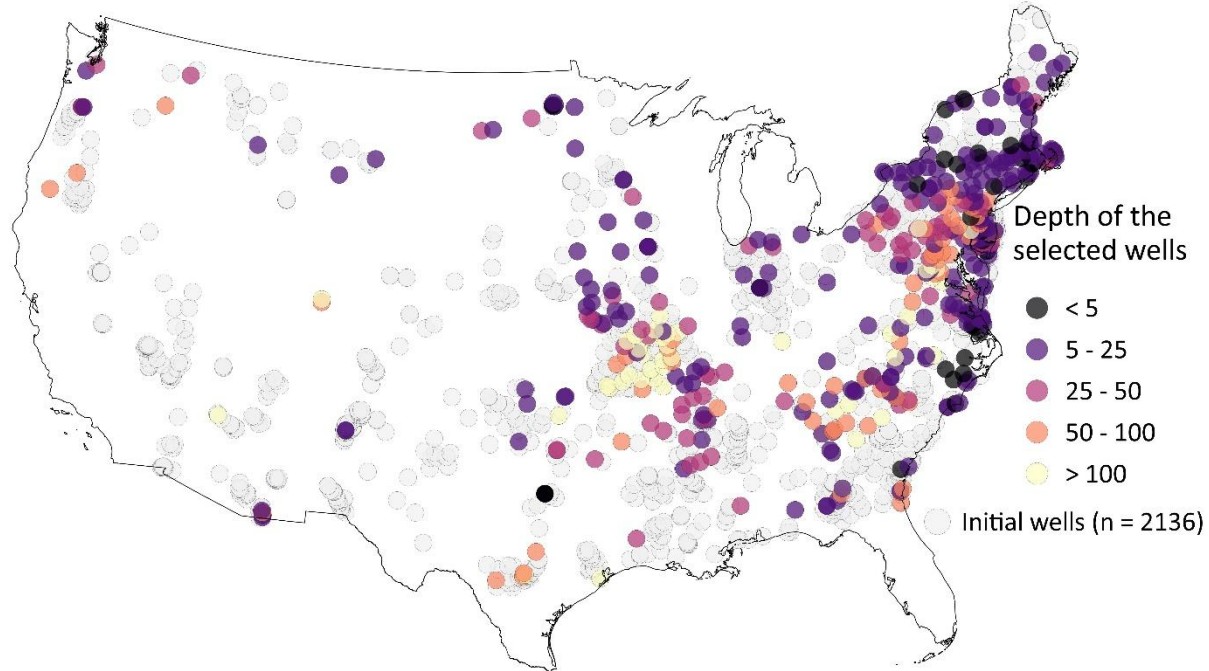

**Figure 2: Location of all the available observation wells with (> 2 years) daily data and the final selected wells (shown in color) with their screen depths in meters.**

**2.3 Influence of water table depth variations and nearby streams on RpSy**

Although the RpSy dataset may not provide direct estimate of recharge, but it can still capture the temporal consistency of groundwater recharge. This assumes that the specific yield ($S_y$) varies minimally in time, as has been considered in numerous groundwater studies (Maréchal et al., 2006; Gumuła-Kawęcka et al., 2022; Ala-Aho et al., 2015; Gehman et al., 2009; Varni et al., 2013). Under this assumption, temporal variations of recharge and RpSy time series are expected to be the same. However, several studies (Lv et al., 2021; Shah and Ross, 2009) have pointed out the temporally varying nature of specific yield depending on water table depth, time of drainage, and other influencing factors, many of which indirectly affect water table variations and drainage rate. We perform additional analysis to evaluate the role of groundwater depth variations on $S_y$ and, consequently, on RpSy. To this end, time varying $S_y$ is evaluated using the method derived in Crosbie et al. (2005), where a constant ultimate specific yield ($S_{yu}$) is dynamically modified using the equation

$$\frac{S_y}{S_{yu}} = 1 - \frac{1}{\{1+(\alpha d)^n\}^{1-\frac{1}{n}}} \tag{2}$$

Here, d is the groundwater depth. The van Genuchten drainage parameters, $\alpha$ and n are obtained from POLARIS data (Chaney et al., 2019).

We also specifically identify wells where the likelihood of groundwater dynamics, and consequently recharge, being significantly affected by nearby streams is minimal. To this end, we filter out wells that are either farther than a threshold distance from big-enough streams (bankfull width > 5m) or has groundwater depth in it being higher than the river level. The difference in groundwater and river elevations follows the procedure in Jasechko et al. (2021). Notably, Jasechko et al. (2021) considered threshold distances of 250 m and 1 km, though determining distance thresholds to demarcate the influence of streams on groundwater dynamics remains challenging. While using a threshold of 250 m and other criteria, 420 wells are selected (Supplementary text S1, Figure S2, S3, Table S1). Users may consider just these wells for benchmarking. Despite these efforts to ensure good quality data, there is still a possibility of some contamination to recharge derived from the WTF method at the selected sites.

## 3 Data Assessment

Given that the presented dataset is first of its kind in providing an observational data-based recharge equivalent at a daily resolution, a direct validation is not possible. However, a comparison of the presented data series to recharge estimates from USGS is performed to understand the similarities and differences between these two products. Furthermore, to qualitatively test the physical plausibility of the results, the presented data is also assessed vis-à-vis the direct influencing variables, i.e., precipitation and evapotranspiration. But, before performing the aforementioned evaluations, we first inter-compare the RpSy estimates using both constant and variable Sy (i.e., RpSyu). Results indicate a correlation higher than 0.8 between the two methods in 94% of the sites (Figure S4). Given the similarity of the two data sets (Figure S4, S5, Table S2) and considering that using variable Sy requires soil drainage parameters (see Eqn. 2), which are inherently uncertain (Gupta et al., 2022; Dai et al., 2019), we proceed with RpSy obtained from constant Sy for the subsequent analysis.

### 3.1 Comparison against USGS recharge estimates

We compare the long-term RpSy for selected observation wells against the USGS monthly recharge estimates (Reitz et al., 2017a; Reitz and Sanford, 2019a) available at 1 km × 1 km resolution between October 2003 and December 2015 for the Conterminous United States. The USGS dataset has been developed through data analysis on water budget components from

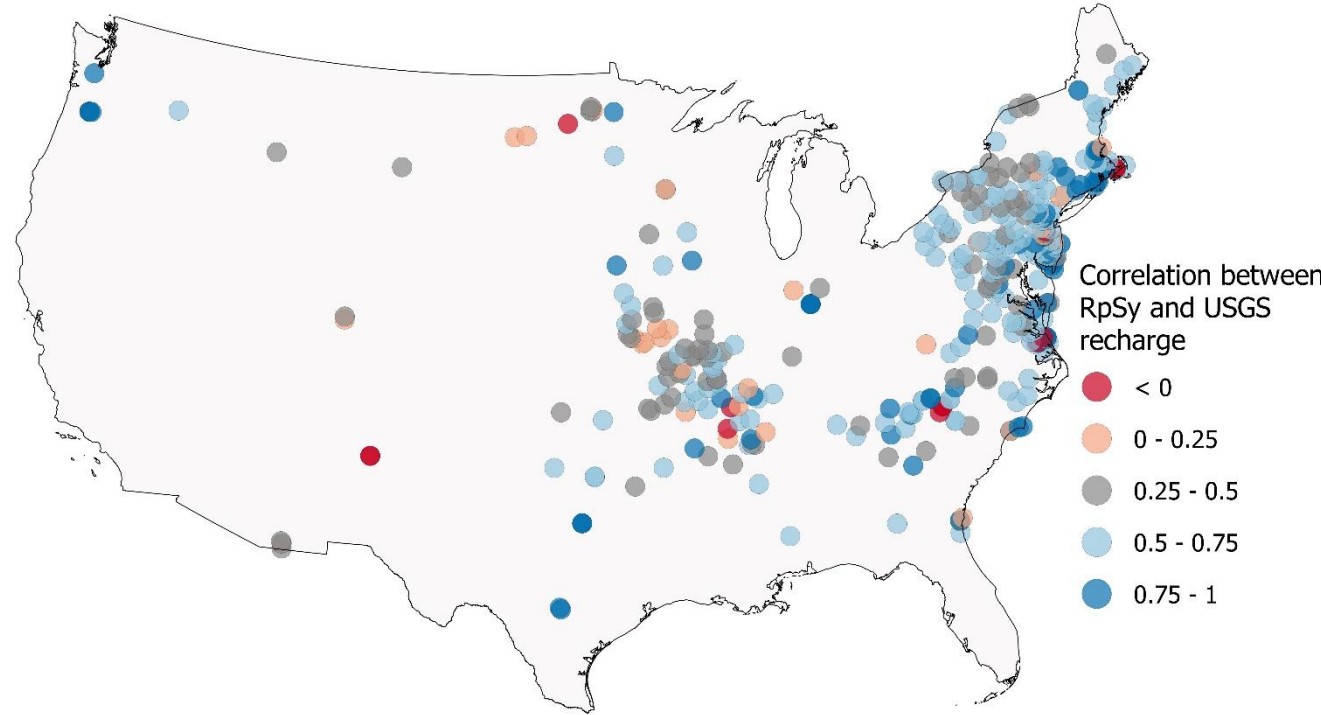

**Figure 3: Spatial variation of temporal correlation between RpSy (m) and USGS recharge (m).**

various sources such as PRISM precipitation (Daly et al., 2008), SNODAS snow water equivalent (Snow Data Assimilation System (SNODAS) Data Products at NSIDC, Version 1 | National Snow and Ice Data Center, 2022), SSEBop-WB evapotranspiration (Reitz et al., 2017b), USGS groundwater-sourced irrigation map (Reitz et al., 2017a) and monthly surface

runoff (Reitz and Sanford, 2019b). Figure 3 highlights the spatial variation of temporal correlations between RpSy and USGS recharge. A plot (Figure S6) of temporal correlation between RpSyu and USGS recharge indicates similar spatial variations. The average spatial Pearson's correlation between RpSy and USGS recharge at the observation wells is R = 0.54, with ~1/5 of the wells having R > 0.75. We find positive correlations in 95% of all the selected wells, with the highest correlation of 0.99 and the lowest correlation of -0.8. Higher correlations are generally observed in the moist northeastern US, while correlations

are lower in the midwestern US. It is conjectured that the relatively poor performance at several wells in the mid-western states is attributable to the uniform distribution of county-scaled groundwater-sourced irrigation water use data in the assessment of USGS recharge (Reitz et al., 2017a). In contrast, wells selected for RpSy derivation are expected to be not influenced by irrigation. The disagreement also could be due to the scale mismatch between the RpSy, which provides recharge equivalent estimates at a point scale (near the wells), and the gridded USGS product. Biases in precipitation and evapotranspiration data

used in the USGS product can also contribute to this disagreement. An additional source of mismatch could be linked to the USGS product neglecting the change of storage in the water balance approach and the inherent uncertainties in their model structure (Reitz et al., 2017a). It is to be noted that RpSy essentially captures the recharge flux reaching the groundwater table, while the recharge estimate in USGS product's is the water flux leaking below the root zone. These two fluxes can be different, especially in settings where significant moisture deficit exists in the soil column between the root zone and the groundwater

table.

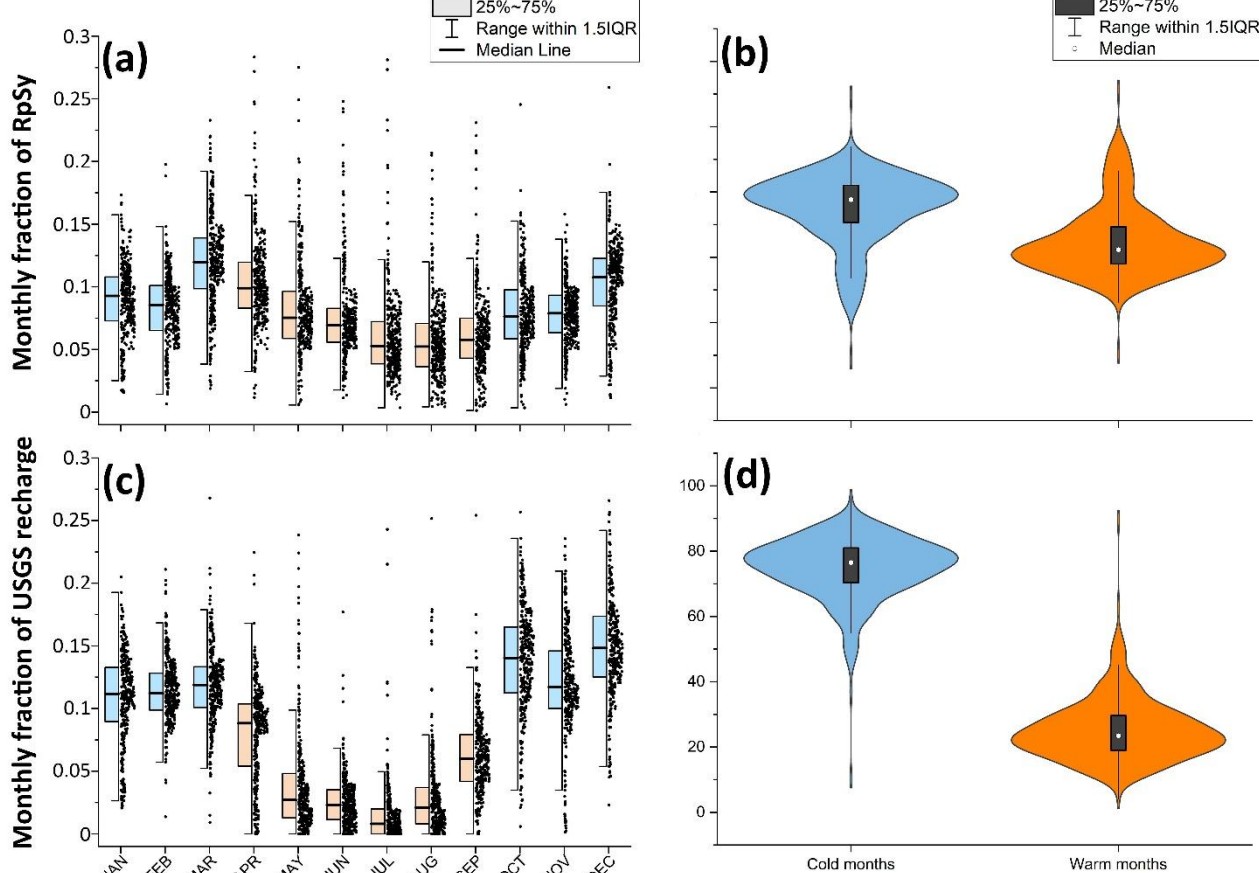

**Figure 4: Fraction of recharge in different months and seasons (i.e., Cold seasons (Oct to Mar), Warm-season (Apr to Sept)) relative to the total recharge(/equivalents) for Recharge per unit specific yield (RpSy) (top, a and b) and USGS (bottom, c, and d) recharge products. In this plot, USGS recharge data for the grids with RpSy estimates are used. IQR indicates the interquartile range.**

To investigate the temporal agreement of the two products, we calculate the fraction of recharge in different months and seasons (i.e., colder months (Oct to Mar) and warmer months (Apr to Sept)) relative to the total recharge equivalent in the respective dataset. The temporal recharge fraction suggests that USGS and RpSy show larger fractions in cold months. The USGS product shows a higher recharge fraction in cold months than RpSy, resulting in a lower fraction in warm months. The lowest monthly fraction is observed around July for USGS recharge and in August for RpSy. The USGS recharge also seems to peak earlier (in December) than RpSy (in March) (Figure 4).

### 3.2 RpSy variations with Precipitation (Ppt) and Evapotranspiration (ET)

Next, we assess the inter-annual and intra-annual variations (Table S3) in RpSy vis-à-vis Ppt and (Ppt-ET). Evaluation of inter-annual variations includes comparisons of normalized annual recharge fraction (normRpSy, henceforth) with normalized annual precipitation (normPpt, henceforth) or normalized annual precipitation minus evapotranspiration (normPpt-ET, henceforth). Here, the normalized variables (e.g., normRpSy) are calculated as the ratio of its annual magnitude for each year and the overall mean over all the data years. A plot of the normRpSy and normPpt in Figures 5a and S7a show that the inter-annual variation of normRpSy is much larger than precipitation fractions. This points to the ratio of recharge to precipitation being generally much higher in wetter years than in drier years. In contrast, the inter-annual variation of normRpSy is relatively muted with respect to (Ppt-ET), highlighting that the ratio of recharge to (Ppt-ET) is relatively smaller in wetter years than in drier years (Figures 5b and S7b). A similar spatial plot (Figure S8) showing the inter-annual variation of normalized annual recharge (normRpSyu), precipitation (normPpt), and Ppt-ET (normPpt-ET) has been shown in the supplementary section.

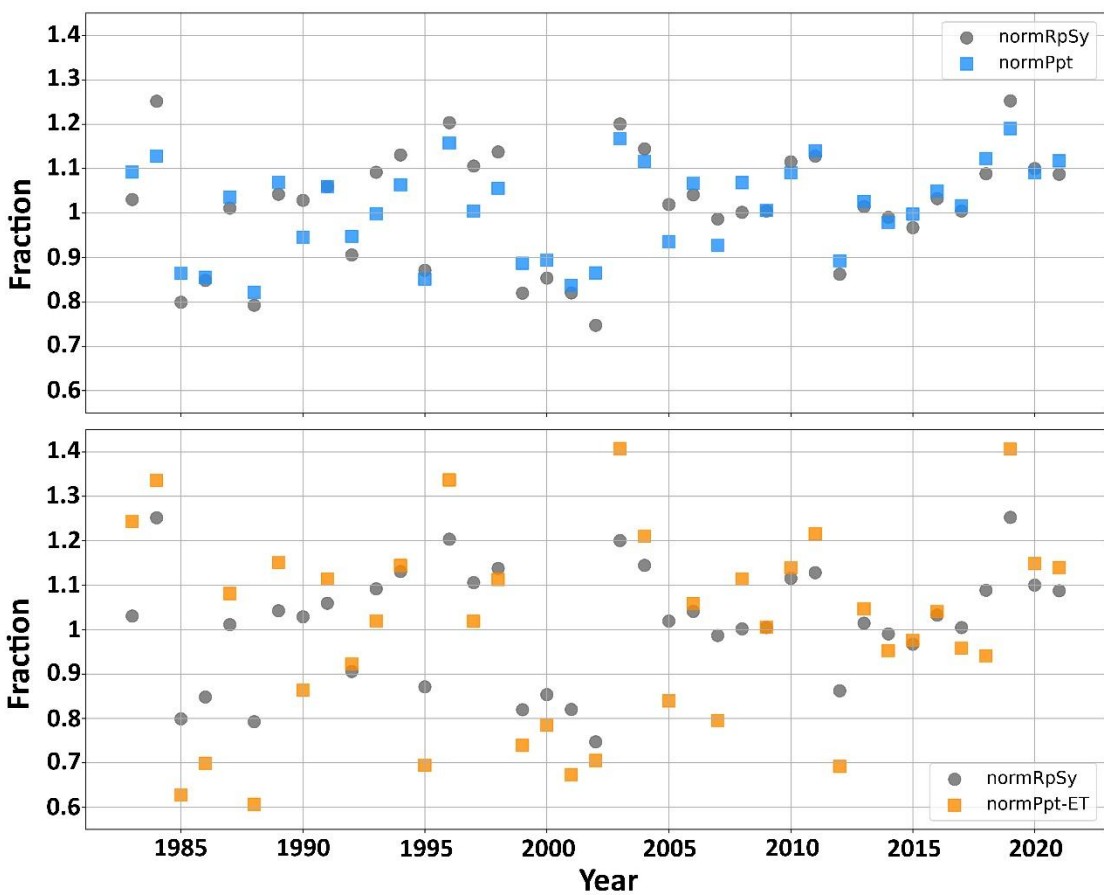

**Figure 5: Inter-annual variation of normalized annual recharge (normRpSy, shown using grey solid dots), precipitation (normPpt, blue squares), and Ppt-ET (normPpt-ET, orange squares).**

To assess the intra-annual variation of RpSy vis-à-vis Ppt and Ppt-ET, we evaluate the centroidal date, defined as the day of the water year corresponding to the center of mass of the daily mean time series averaged over multiple water years (a

245 water year ranges from 1 October to 30 September). The centroidal date is calculated by first obtaining the mean cumulative daily time series of the variables (i.e., RpSy, Ppt, Ppt-ET) across all considered water years and then identifying the date on which the cumulative value is half of the water year total. We find that centroidal date for (Ppt-ET) < centroidal date of RpSy < centroidal date of Ppt (Figure 6, S9). The centroidal date of (Ppt-ET) < centroidal date of RpSy is likely due to a larger runoff ratio during the winter period. For example, for a shallow well in New Jersey, we notice centroidal date of (Ppt-ET) on 4

February, which is relatively earlier than the centroidal date of RpSy and Ppt, which are on 24 March and 5 April, respectively (Figure 6d). It is to be noted that an overestimation in ET (especially during winter) can lead to (Ppt-ET) centroidal date being earlier as well. The centroidal date of RpSy < centroidal date of Ppt is because of larger ET losses in summer.

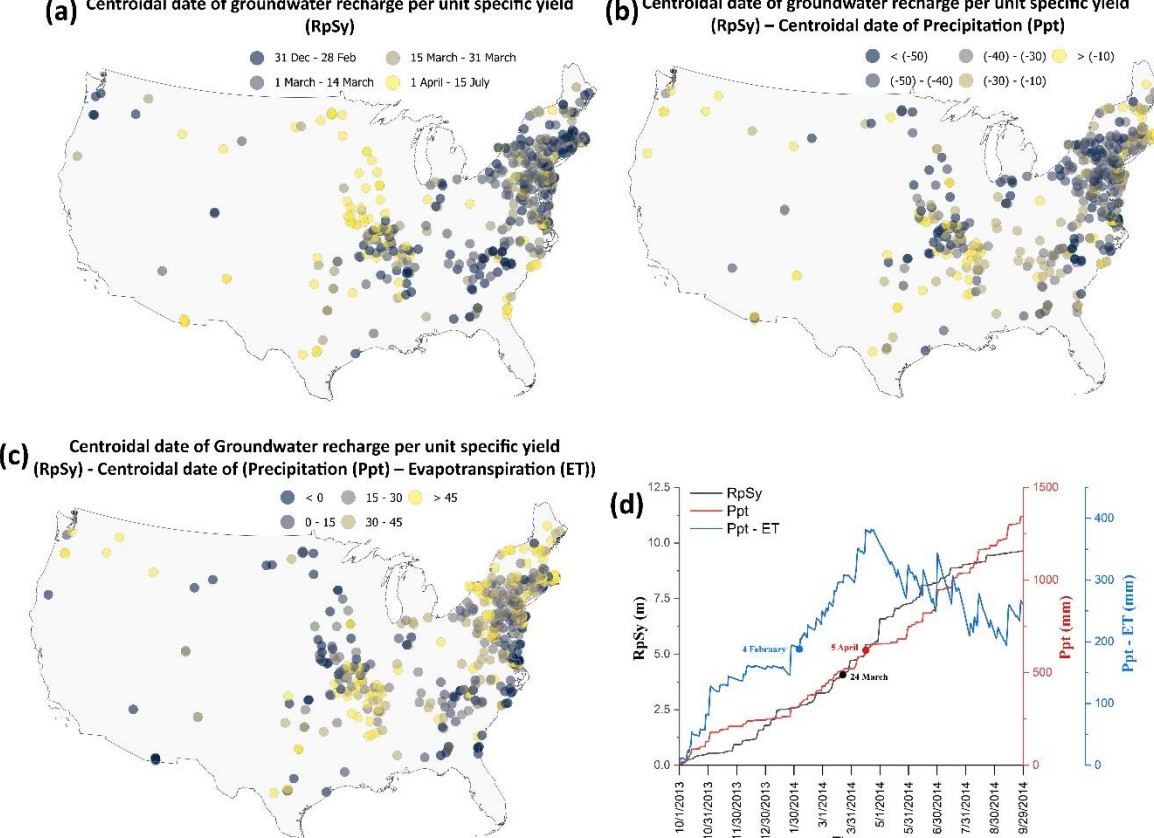

**Figure 6: Centroidal date for RpSy (a). Distance between centroidal date of RpSy and Ppt (b), and (Ppt-ET) (c). A negative (positive) value indicates a later (earlier) centroidal date for the variable with respect to that of RpSy. Day 1 is the start of the water year, i.e., Oct. 1. Also shown in d is a representative example of cumulative time series and corresponding estimated centroidal dates for a shallow well (Well ID: 400232074213201) in New Jersey.**

## 4 Data availability

The final product we provide is continuous daily RpSy time series data estimated using the WTF method for all 485 observation wells (Figure 2). Additionally, we also provide data of daily RpSy time series (RpSyu) while considering a groundwater depth-dependent $S_y$. Furthermore, we also identify the observation wells that are less likely to be affected by water leakage from nearby streams based on the criteria outlined in section 2.3. These data files, provided in the .csv format, consist of three columns for each observation well, with the first column listing the local time while the second and third columns provide the RpSy and RpSyu time series, respectively. The file containing site information for all the selected wells has four columns that detail the USGS ID of the groundwater well, its latitude (Lat), longitude (Long), and screen depth (depth) (Figure S10). We also provide the van Genuchten drainage parameters (i.e., α and n) used to derive RpSyu and nearest stream attributes used to evaluate nearby streams' impact for each observation well. The data file can be opened in most text editors and spreadsheets. The data can be found online at Zenodo: https://doi.org/10.5281/zenodo.13323242 (Malakar et al., 2024).

## 5 Code Availability

We performed all steps, including data processing, MRC analysis, and RpSy calculations in R; and RpSyu and nearest stream attributes calculations in Python. All the codes, along with sample input datasets, can be found at the following link: https://doi.org/10.5281/zenodo.13323242 (Malakar et al., 2024).

**6 Limitations**

RpSy estimate from WTF method can be affected by temporal variation in $S_y$. Although a sincere effort has been made to
explicitly account for the influence of water table depth on $S_y$ and consequently on RpSy (see section 2.3), influences on $S_y$
from other factors (Lv et al., 2021) remain unaccounted for. RpSy evaluation using the WTF method can also be influenced
by several other factors, including stage variations in a nearby surface water body, barometric pressure shifts, abstraction,
groundwater pumping in the vicinity, and lateral flow contributions from uplands. As mentioned in section 2.2, only sites with
a good MRC curve fit are used. This ensures the applicability of the WTF method and helps filter out sites with significant
pumping impacts. However, the method is not foolproof and cannot completely guarantee the absence of groundwater pumping
effects. We also identify a smaller subset of sites where the contribution to recharge from nearby surface water bodies is
expected to be minimal, but the methodology cannot ensure zero contribution. Additionally, we quantify RpSyu or recharge
per specific yield that considers a time-varying specific yield. In instances where specific yield fluctuates temporally due to
precipitation induced variations in groundwater table depth, RpSyu is expected to be more effective in capturing daily recharge
fluctuations. This is particularly relevant in regions with a shallow groundwater table or in soils with fine textures, such as
clayey soils, which have a large capillary fringe. In these conditions, the specific yield is significantly reduced. Part of the
reason is that the capillary fringe retains water tightly thereby reducing the freely drainable portion of water. Also, when the
groundwater is shallow, the unsaturated zone above the capillary fringe is either minimal or absent. As a result, the soil's ability
to release water is constrained. These conditions could be common in regions experiencing large fluctuations in water table
depth, such as areas with large season precipitation, intensive irrigation, or heavy groundwater pumping. However, since RpSy
and RpSyu are the first of their kind to provide observational data-based recharge equivalents at a daily resolution, direct
validation is not feasible. The scale mismatch between RpSyu and USGS recharge data, and inherent assumptions in the USGS
product also preclude a direct one-to-one comparison.

Recharge estimate from WTF method may also be affected by Lisse effect, which is characterized by large water level rise
than expected for an infiltration amount due to trapped air in the unsaturated zone following a sudden high-intensity rain event
(Crosbie et al., 2005; Cuthbert, 2010). Since Lisse effect occurs under very specific conditions of intense rainfall, fine-textured
soil, and shallow groundwater table, it is expected to be less frequent. Irrespectively, as there is no sure shot method for
detecting and eliminating the Lisse effect from groundwater levels without using additional data on subsurface moisture or
potential, which are generally unavailable at most groundwater well sites, some influence of the Lisse effect may still be
present in the derived RpSy data.

**7 Conclusions**

The study presents a novel benchmark dataset of groundwater recharge per unit specific yield (RpSy) at daily temporal
resolution. The estimates are based on WTF and MRC methods and only require readily available precipitation and GWL data
to obtain the recharge equivalents. The output data (i.e., RpSy) are available for download from the zenodo.com platform
(Malakar et al., 2024). The datasets for continuous daily RpSy, well locations, and time are provided for end-user use. Given
that the presented data set is first of its kind in providing an observational data-based recharge equivalent at a daily resolution,
a direct validation is not possible. However, a comparison of the presented data series to recharge estimates from USGS is
performed. To qualitatively assess the physical plausibility of the results, the presented data is also studied vis-à-vis the direct
influencing variables, i.e., precipitation and evapotranspiration. While there is no available dataset that demonstrates the actual
recharge time series at daily resolution, a comparison of the presented data series to monthly recharge estimates from USGS
is performed for users of the data. Further analyses highlight that the RpSy product shows physically plausible temporal
variations vis-à-vis the variations in Ppt and ET at both inter- and intra-annual scales.

While the RpSy data does not offer direct recharge estimates, it still captures the variations and changes in groundwater recharge over time at daily to coarser temporal resolution. Hence, despite the limitations, uncertainty, and associated caveats discussed in section 2 and 6, the RpSy dataset can be used to validate temporal consistency of recharge estimates derived from empirical methods (Reitz et al., 2017a; Reitz and Sanford, 2019a), physically-based land surface models (Anurag and Ng, 2022; Li et al., 2021; Niraula et al., 2017) or integrated hydrologic models (Kumar and Duffy, 2015; Kollet and Maxwell, 2006; Kumar et al., 2009; Therrien et al., 2010). The RpSy dataset can be utilized for analysing the timing, frequency, and duration of recharge events. Since RpSy provides fluctuations at a daily scale, researchers can use the temporal patterns to assess whether the abovementioned models have the ability to accurately simulate groundwater recharge variability. The match or mismatch between the temporal alignment of RpSy based recharge and model based recharge outputs can provide an assessment of the model's capability to replicate the event based response to hydroclimatic forcings. Furthermore, the data may also be used to validate the functional relationship between recharge and associated factors as represented in land surface and global hydrologic models. Gnann et al. (2023) demonstrated that theoretical and empirically based functional relationships for recharge differ significantly from global water models. Even when a model produces highly accurate predictions, it may still poorly simulate the strength of functional process couplings. In other words, it may produce right results for the wrong reasons. Such models are likely to underperform during periods when the forcing characteristics are different than those in the training data. The derived benchmark RpSy data, along with forcing variables, can be used to validate the functional relationships represented in models of recharge, using one of the several diagnostic methods such as information theory (Ruddell et al., 2019), causality mapping (Barnett and Seth, 2014; Runge et al., 2019; Runge, 2018), and convergence cross mapping (Ye et al., 2015). The RpSy data may also be used to temporally downscale long-term recharge estimates from observations, thus facilitating generation of recharge inputs for groundwater models (Kim et al., 2008). In circumstances where high confidence specific yield values are available and/or obtainable from field measurements, hydrogeological surveys, or from literature, the RpSy data can be converted into recharge estimates (Recharge = RpSy $\times$ Sy). In these cases, a direct comparison can be made between the magnitude of modelled recharge and RpSy based recharge. Additionally, the data may be used for an improved understanding of the role of different forcing and antecedent hydrologic conditions on groundwater recharge and, thus helping manage groundwater aquifers under water-stress conditions.

**Author contribution**

MK conceived the study and provided project administration and supervision. MK and PC acquired funding to support the project. PM compiled the data, developed relevant codes for analyses and visualizations, and generated results. AA developed and implemented code to obtain modified RpSy that accounts for the influence of water table depth on $S_y$. AA also identified the gauging stations with smaller likelihood of influence on groundwater dynamics from nearby streams. PM and MK designed the methodology, performed data analyses, and drafted the manuscript. PC and AT provided thoughtful inputs. All authors edited the manuscript and helped improve it.

**Competing interests**

The authors declare no competing interests.

**Acknowledgments**

Groundwater level (GWL) records for the last four decades (1983-2022) are retrieved from the US Geological Survey's (USGS) National Water Information System web interface (link: https://waterdata.usgs.gov/nwis/gw). Daily Precipitation data from

PRISM (Daly et al., 2008; data link: https://prism.oregonstate.edu/) are used. Evapotranspiration data is obtained from the Global Land Evaporation Amsterdam Model (GLEAM) (Martens et al., 2017; Miralles et al., 2011; data link: https://www.gleam.eu/).

**Financial support**

This work was supported by the NSF OIA-2019561 and NOAA NA22NWS4320003 grants.

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
