# Peer review of "An in-situ daily dataset for benchmarking temporal variability of groundwater recharge"

_Earth System Science Data, 2024_

## Author Comment (AC1)

**Legend**

Reviewers' comments

Authors' responses

Direct quotes from the revised manuscript

**Reply to Reviewers' comments (Reviewer#1)**

Reviewer #1: This is the review for the paper titled "An in-situ daily dataset for benchmarking temporal variability of groundwater recharge" by Malakar et al. The authors estimated groundwater recharge per unit specific yield (RpSy) at the 485 groundwater monitoring wells across the United States. They adopted the water table fluctuation method on the daily groundwater table time-series. Here are my major comments:

I like the way the authors represented groundwater monitoring well data in this manuscript. They performed analysis using daily time-scale data.

Response: We thank Dr. Bhanja for his time reviewing our manuscript and providing detailed comments and suggestions. Point-by-point replies to the comments or suggestions made can be found below.

Reviewer #1: The authors mentioned that the RpSy concept was introduced to reduce errors associated with the recharge estimation arising from the uncertainties in the specific yield (Sy) data. RpSy is nothing but the dh – the change in water table. I think this may not provide the representative values of change in recharge rates. For example, groundwater recharge signature in alluvium will differ a lot from the hard rock areas with a similar change in water table. I think the authors can reduce emphasizing RpSy as a central point of this manuscript, rather, focus more on creating the unique database.

Response: We thank Dr. Bhanja for this comment. In this manuscript, our objective is to benchmark temporal variations at gauging locations, not to spatially compare recharge variations using RpSy. We would like to underscore that estimation of RpSy cannot merely be considered as the change in water table (dh). After a recharge event, the groundwater table may rise and then recede. Calculating recharge simply by taking the dh between the start of the recharge event and a point further along the recession period could yield a small or even negative recharge estimate. In our approach to calculating recharge using the Water Table Fluctuation (WTF) method, we determine the recharge by evaluating ΔH, which is the difference in water table height between two consecutive time steps adjusting for what the height at the current time step would be if it were receding according to the rate defined by the master recession curve (MRC). Notably, the time steps is 1 day in our case, which is much shorter than the duration of a recharge event. An MRC predicts the characteristic rate of change of water-table level as a function of the H. To summarise, ΔH is not simply the vertical distance

between two temporal points; rather, it incorporates the characteristic rate of change of the water table level in its calculation. Thus, our aim is to build this unique RpSy database; even without incorporating Sy, the data presented remains valuable.

Following the reviewer's comment, we clarified this in the text:

It is to be noted that the WTF based groundwater recharge incorporating MRC is not merely the difference in water table height between two time points. For instance, after a recharge event, the groundwater table may rise and then recede. Calculating recharge simply by taking the difference in water table height between the start of the recharge event and a point further along the recession period could yield a small or even negative recharge estimate. In our approach to calculating recharge using the WTF method, we determine the recharge by evaluating the difference in groundwater table height between the current and next time step, adjusting for what the height at the current time step would be if it were receding according to the rate defined by the MRC. This variable, $\Delta H$, is evaluated between two consecutive time steps, 1 day in our case. Notably, the time step is much shorter than the duration of a recharge event. **[Page: 3, Line: 100-108]**

Reviewer #1: Section 3.1 and Figure 3: Correlation between RpSy and USGS-based groundwater recharge show less than 0.5 values across the majority of the mid-western, dryland areas. I understand USGS-based recharge estimates using the water budget approach, can the mismatch show bias in precipitation or any other data? The magnitude mismatch is understandable, the patterns should match unless uncertainties present in the data. Authors may consider using other recharge data for comparison in those areas.

Response: Thank you for your insightful comment regarding the correlation between RpSy and USGS-based groundwater recharge in the mid-western, dryland areas, particularly to potential biases in precipitation or other data sources.

We agree that the observed correlation values below 0.5 in these regions call for further scrutiny. One likely explanation for the mismatch could be the inherent differences in the methodologies used to derive RpSy and USGS-based recharge. While the USGS-based recharge estimates employ a water-budget approach, which primarily reflects precipitation and surface runoff inputs at a coarser resolution, the RpSy data are derived from the water table fluctuation (WTF) method, which captures localized recharge responses at a point-scale. This scale mismatch can lead to differing recharge estimates, especially in regions with significant heterogeneity in soil moisture, precipitation patterns, and land use.

Additionally, it is conjectured that the relatively poor performance at several wells in the mid-western states is attributable to the uniform distribution of county-scaled groundwater-sourced irrigation water use data in the assessment of USGS recharge (Reitz et al., 2017a). In contrast, wells selected for RpSy derivation are expected to be not influenced by irrigation. An additional source of mismatch could be linked to the USGS product neglecting the change of storage in the water balance approach and the inherent uncertainties in their model structure (Reitz et al., 2017a). It is to be noted that RpSy essentially captures the recharge flux reaching the groundwater table, while the recharge estimate in USGS product's is the water flux leaking below the root zone. These two fluxes can be different,

especially in settings where significant moisture deficit exists in the soil column between the root zone and the groundwater table.

As you rightly pointed out, biases in precipitation data may contribute to the observed differences in patterns. For example, the USGS dataset reflects precipitation data at a gridded resolution, while the RpSy estimates reflect local recharge conditions that may be more sensitive to spatial and temporal variability in rainfall. In dryland areas, where precipitation events are often sporadic and intense, such differences in temporal and spatial resolution could significantly impact recharge estimates. In fact, there could be biases in the USGS data from ET estimates used therein as well.

We also acknowledge the suggestion to explore other recharge datasets for comparison. Notably, the current analysis focuses on the developed benchmark data, which is first of its kind in providing an observational data-based recharge equivalent at a daily resolution. The comparison with the USGS product is not a direct validation but rather done to highlight the alignment and disagreement with an established product. We have noted that in the conclusion section, that the RpSy dataset can be used to validate temporal consistency of other recharge estimates, derived from empirical methods, physically-based land surface models, or integrated hydrologic models.

Following the reviewer's concern, we have modified the existing text in the manuscript:

It is conjectured that the relatively poor performance at several wells in the mid-western states is attributable to the uniform distribution of county-scaled groundwater-sourced irrigation water use data in the assessment of USGS recharge (Reitz et al., 2017a). In contrast, wells selected for RpSy derivation are expected to be not influenced by irrigation. The disagreement also could be due to the scale mismatch between the RpSy, which provides recharge equivalent estimates at a point scale (near the wells), and the gridded USGS product. Biases in precipitation and evapotranspiration data used in the USGS product can also contribute to this disagreement. An additional source of mismatch could be linked to the USGS product neglecting the change of storage in the water balance approach and the inherent uncertainties in their model structure (Reitz et al., 2017a). It is to be noted that RpSy essentially captures the recharge flux reaching the groundwater table, while the recharge estimate in USGS product's is the water flux leaking below the root zone. These two fluxes can be different, especially in settings where significant moisture deficit exists in the soil column between the root zone and the groundwater table. **[Page: 7, Line: 203-113]**

**Reference:**

Reitz, M., Sanford, W. E., Senay, G. B., and Cazenas, J.: Annual Estimates of Recharge, Quick-Flow Runoff, and Evapotranspiration for the Contiguous U.S. Using Empirical Regression Equations, J. Am. Water Resour. Assoc., 53, 961–983, https://doi.org/10.1111/1752-1688.12546, 2017a.

Reviewer #1: Figure 4a and 4c look similar to me and they are not reflecting the patterns observed in Figure 4b and 4d. Please revisit the figures.

Response: We thank Dr. Bhanja for this perceptive note. In response to the reviewer's comment, we acknowledge that during the production of the combined figure, we mistakenly pasted identical Figures in 4a and 4c, which indeed resulted in a lack of alignment with the patterns observed in Figures 4b and 4d. We sincerely appreciate the reviewer for pointing out this oversight. Following your

feedback, we have corrected the figure accordingly. The corrected Figure 4 is incorporated in the manuscript.

[Figure]

**Figure 4: Fraction of recharge in different months and seasons (i.e., Cold seasons (Oct to Mar), Warm-season (Apr to Sept)) relative to the total recharge(/equivalents) for RpSy (top, a and b) and USGS (bottom, c, and d) recharge products. In this plot, USGS recharge data for the grids with RpSy estimates are used. IQR indicates the interquartile range.**

---

## Author Comment (AC2)

**Reply to Reviewers' comments (Reviewer#2)**

Reviewer #2: This paper provides a useful dataset of estimated daily recharge per unit specific yield (RpSy) across 485 well locations in the US derived using the water table fluctuation method on daily groundwater table time series. Overall the paper seems to be a useful contribution, but I suggest that the following aspects are further considered before publication:

Response: We would like to thank the reviewer for their positive feedback and for recognizing the value of our dataset. We have carefully considered the suggestions made by the reviewer, and incorporated them to the best of our ability.

Reviewer #2: To understand interannual variations in recharge (Fig. 5) would it be useful to not just consider a timeseries plot, but also make scatter plots of drivers (PPpt, or Ppt-ET) and responses (recharge). This may help to better understand and quantify their linkages.

Response: We would like to thank the reviewer for their insightful suggestion regarding the analysis of interannual variations in recharge. Following the reviewer's suggestion, we have added the scatter plots in the supplementary information to illustrate the relationship between key drivers (e.g., Ppt or Ppt-ET) and RpSy, in addition to the timeseries plot. These new visualizations help to clarify and quantify the linkages between drivers and responses. Thank you for helping us improve the clarity of our analysis. We included in the supplementary file,

A plot of the normRpSy and normPpt in Figures 5a and S7a show that the inter-annual variation of normRpSy is much larger than precipitation fractions. This points to the ratio of recharge to precipitation being generally much higher in wetter years than in drier years. In contrast, the inter-annual variation of normRpSy is relatively muted with respect to (Ppt-ET), highlighting that the ratio of recharge to (Ppt-ET) is relatively smaller in wetter years than in drier years (Figures 5b and S7b). **[Page: 8, Line: 229-134]**

[Figure]

**Figure S7: Scatter plot showing the variation between (a) normalized annual recharge (normRpSy) and precipitation (normPpt) shown using blue dots; (b) normalized annual recharge (normRpSy) and Ppt-ET (normPpt-ET) shown using orange dots.**

Reviewer #2: In figure 4, panel a and d look completely identical but should be different. Check if an presentation error is made here.

Response: We appreciate the reviewer's note regarding Figure 4. Upon review, it was found that we inadvertently made a mistake during the production of the combined figure, by pasting identical figures in panels 4a and 4c. We are grateful to the reviewer for pointing this out, and we have since corrected the figure accordingly.

[Figure]

**Figure 4: Fraction of recharge in different months and seasons (i.e., Cold seasons (Oct to Mar), Warm-season (Apr to Sept)) relative to the total recharge(/equivalents) for RpSy (top, a and b) and USGS (bottom, c, and d) recharge products. In this plot, USGS recharge data for the grids with RpSy estimates are used. IQR indicates the interquartile range.**

Reviewer #2: the quality of all figures rather limited. For example, Figure 1: make the timeseries somewhat more readable (using a highjer resolution figure output and a slight change of line styles may help here. Please check all figure to potentially up the standard.

Response: We appreciate the feedback on figure quality. We have enhanced Figure 1 by enhancing the resolution and adjusting line styles to improve readability. Additionally, we have reviewed and made similar improvements across all figures (Figure 2, Figure 3, Figure 5, Figure 6) to meet higher standards.

[Figure]

**Figure 1: (a) Schematic representation of ΔH evaluation in the WTF method, which is then used to obtain daily groundwater recharge using Eqn 1, (b) a typical dH/dt vs. H plot used to derive the MRC. The hydrographs (c) and (e) show daily groundwater level (GWL), precipitation (Ppt), and estimated recharge per unit specific yield (RpSy, discussed later) for a selected and a rejected well,**

**respectively. Here, selection/rejection is based on representativeness of the MRC, which here is determined based on an adj-R2 value of 0.2. (d) and (f) show the dH/dt vs. H plot for the corresponding selected and rejected wells, respectively.**

[Figure]

**Figure 2: Location of all the available observation wells with (> 2 years) daily data and the final selected wells (shown in color) with their screen depths in meters.**

[Figure]

**Figure 3: Spatial variation of temporal correlation between RpSy and USGS recharge.**

[Figure]

**Figure 5: Inter-annual variation of normalized annual recharge (normRpSy, shown using grey solid dots), precipitation (normPpt, blue squares), and Ppt-ET (normPpt-ET, orange squares).**

Reviewer #2: For the comparison of center of mass, please make more direct comparison of these datapoints than just maps. Also note that a "centroid" does not match "the date on which the cumulative value is half of the yearly total in a water year". A centroid represents the mean of a cumulative (recharge) distribution whereas the textual description would represent the date of that matches the median.

Response: For better clarity, in response to the reviewer's comment, we have edited the relevant text. Additionally, we added Fig. S9 that shows a direct comparison of centroidal dates for RpSy, Ppt, and Ppt-ET.

To assess the intra-annual variation of RpSy vis-à-vis Ppt and Ppt-ET, we evaluate the centroidal date, defined as the day of the water year corresponding to the center of mass of the daily mean time series averaged over multiple water years (a water year ranges from 1 October to 30 September). The centroidal date is calculated by first obtaining the mean cumulative daily time series of the variables (i.e., RpSy, Ppt, Ppt-ET) across all considered water years and then identifying the date on which the cumulative value is half of the water year total. We find that centroidal date for (Ppt-ET) < centroidal date of RpSy < centroidal date of Ppt (Figure 6, S9). The centroidal date of (Ppt-ET) < centroidal date of RpSy is likely due to a larger runoff ratio during the winter period. For example, for a shallow well in New Jersey, we notice centroidal date of (Ppt-ET) on 4 February, which is relatively earlier than the centroidal date of RpSy and Ppt, which are on 24 March and 5 April, respectively (Figure 6d). It is to be noted that an overestimation in ET (especially during winter) can lead to (Ppt-ET) centroidal date

being earlier as well. The centroidal date of RpSy < centroidal date of Ppt is because of larger ET losses in summer. **[Page: 10, Line: 238-248].**

[Figure]

**Figure S9: Variation of RpSy centroidal dates in comparison with Precipitation (Ppt) and Precipitation Minus Evapotranspiration (Ppt-ET) centroids for 485 locations. (a) the distribution of centroidal dates for RpSy, Ppt, and Ppt-ET; (b) Histogram of Centroidal Date Differences.**

[Figure]

**Figure 6: Centroidal date for RpSy (a). Distance between centroidal date of RpSy and Ppt (b), and (Ppt-ET) (c). A negative (positive) value indicates a later (earlier) centroidal date for the variable with respect to that of RpSy. Day 1 is the start of the water year, i.e., Oct. 1. Also shown in d is a**

**representative example of cumulative time series and corresponding estimated centroidal dates for a shallow well (Well ID: 400232074213201) in New Jersey.**

Thank you for this valuable feedback, as it has allowed us to improve both our analysis and the clarity of our terminology in the manuscript.

---

## Author Comment (AC3)

**Legend**

Reviewers' comments

Authors' responses

Direct quotes from the revised manuscript

**Reply to Reviewers' comments (Reviewer#3)**

Reviewer #3: This is a review of "An in-situ daily dataset for benchmarking temporal variability of groundwater recharge" by P. Malakar et al. This paper describes the development of a benchmark dataset of groundwater recharge per unit specific yield (RpSy). The authors apply an established Water Table Fluctuation / Master Recession Curve method and QA/QC measures to produce daily groundwater level variation time series data for 485 sites.

As the authors note, there are currently no daily timescale data sets in the literature with which one can compare estimated/modeled results for groundwater recharge. Due to usefulness of the results, I recommend acceptance of the paper after minor revisions.

Response: We thank the reviewer for their thoughtful evaluation. We greatly appreciate the recognition of our efforts in creating a benchmark dataset. We have addressed the suggested minor revisions to further strengthen the manuscript and enhance the readability and utility of our figures and results. Thank you for your recommendation for acceptance following these adjustments.

Reviewer #3: My main request is that more analysis and discussion be given to the RpSyu data, which is the version of the data that includes time varying specific yield, and which is provided in this data set alongside the RpSy data. The authors state that because the correlation between these two data sets is greater than 0.8, they will not include RpSyu data in the comparisons with USGS data, Ppt, ET, etc. It is not surprising that RpSy and RpSyu would generally correlate with each other, but without a comparison between RpSyu data and the USGS data, a user cannot determine whether the additional complexity of the RpSyu calculation adds any value. Some analysis here would help the user decide whether to use the provided RpSy or RpSyu. At a minimum, the map of R2 between the USGS data and the RpSy data should have an equivalent map for RpSyu, and there should be some summary statistics that help readers understand whether RpSy or RpSyu more closely matches the temporal variation in recharge.

Response: We would like to thank the reviewer for highlighting the importance of comparing RpSyu and RpSy data with USGS recharge data to assess the additional complexity's value in RpSyu. Following the reviewer's suggestion, we have generated a spatial map of the correlation between RpSyu and USGS recharge and included summary statistics and distribution plots of correlations for RpSy and RpSyu. These analyses allow users to assess the utility of using RpSyu vs. RpSy. The average correlation between RpSyu and USGS recharge is comparable to that of RpSy. Overall, the results suggest that

while the two datasets are closely related, differences exist. This additional information will aids users in selecting the dataset best suited for their needs.

[Figure]

**Figure S5: Comparison of RpSy and RpSyu correlations with USGS recharge data. (a) the correlations spread for both RpSy and RpSyu, with the orange line indicating the median correlation; (b) histograms show the correlation distributions for RpSy and RpSyu with USGS recharge.**

We also modified and added in the text,

Additionally, we quantify RpSyu or recharge per specific yield that considers a time-varying specific yield. In instances where specific yield fluctuates temporally due to precipitation induced variations in groundwater table depth, RpSyu is expected to be more effective in capturing daily recharge fluctuations. This is particularly relevant in regions with a shallow groundwater table or in soils with fine textures, such as clayey soils, which have a large capillary fringe. In these conditions, the specific yield is significantly reduced. Part of the reason is that the capillary fringe retains water tightly thereby reducing the freely drainable portion of water. Also, when the groundwater is shallow, the unsaturated zone above the capillary fringe is either minimal or absent. As a result, the soil's ability to release water is constrained. These conditions could be common in regions experiencing large fluctuations in water table depth, such as areas with large season precipitation, intensive irrigation, or heavy groundwater pumping. However, since RpSy and RpSyu are the first of their kind to provide observational data-based recharge equivalents at a daily resolution, direct validation is not feasible. The scale mismatch between RpSyu and USGS recharge data, and inherent assumptions in the USGS product also preclude a direct one-to-one comparison. **[Page: 11, Line: 277-288]**

**Table S2: Summary Statistics for Correlation between RpSyu, RpSy, and USGS recharge.**

| | Correlation_RpSyu | Correlation_RpSy |
|---|---|---|
| mean | 0.531049 | 0.541389 |
| standard deviation | 0.256786 | 0.251975 |
| minimum | -0.29906 | -0.8052 |
| 5% (first quartile) | 0.376744 | 0.391686 |
| 50% (median or second quartile) | 0.58046 | 0.582712 |
| 75% (third quartile) | 0.719417 | 0.730783 |

| | | |
|---|---|---|
| maximum | 0.999441 | 0.999503 |

[Figure]

**Figure S6: Spatial variation of temporal correlation between RpSyu and USGS recharge.**

Reviewer #3 (Other comment 1): The interannual variation data in Figure 5 should be presented in table form, showing the R2 between the data sets (and including columns for the RpSyu data).

Response: We thank the reviewer for the comment. Following the reviewer's comment, we have added the table.

**Table S3. Inter-annual variation of normalized annual recharge (normRpSy), precipitation (normPpt), and Ppt-ET (normPpt-ET).**

| Year | RpSy | RpSyu | Ppt | Ppt-ET | $R^2$ for Rpsy | $R^2$ for Rpsyu |
|---|---|---|---|---|---|---|
| 1983 | 103.0 | 103.1 | 109.2 | 124.3 | **RpSy Vs. Ppt** | **RpSyu Vs. Ppt** |
| 1984 | 125.2 | 92.9 | 112.8 | 133.5 | 0.839 | 0.029 |
| 1985 | 79.9 | 108.4 | 86.4 | 62.8 | **RpSy Vs. Ppt-ET** | **RpSyu Vs. Ppt-ET** |
| 1986 | 84.8 | 103.4 | 85.5 | 69.9 | 0.837 | 0.026 |
| 1987 | 101.1 | 106.8 | 103.6 | 108.1 | | |
| 1988 | 79.2 | 107.8 | 82.2 | 60.7 | | |
| 1989 | 104.2 | 98.6 | 106.9 | 115.1 | | |
| 1990 | 102.9 | 106.5 | 94.6 | 86.3 | | |
| 1991 | 105.9 | 117.0 | 106.0 | 111.4 | | |
| 1992 | 90.6 | 102.7 | 94.8 | 92.2 | | |
| 1993 | 109.2 | 80.8 | 99.8 | 101.9 | | |
| 1994 | 113.1 | 94.9 | 106.3 | 114.4 | | |

| | | | | |
|---|---|---|---|---|
| 1995 | 87.1 | 102.9 | 85.1 | 69.4 |
| 1996 | 120.3 | 89.4 | 115.8 | 133.7 |
| 1997 | 110.6 | 120.6 | 100.4 | 101.9 |
| 1998 | 113.8 | 106.8 | 105.6 | 111.2 |
| 1999 | 82.0 | 99.9 | 88.7 | 73.9 |
| 2000 | 85.3 | 104.0 | 89.4 | 78.4 |
| 2001 | 82.0 | 114.9 | 83.7 | 67.4 |
| 2002 | 74.7 | 98.2 | 86.5 | 70.5 |
| 2003 | 120.0 | 118.5 | 116.7 | 140.7 |
| 2004 | 114.4 | 109.0 | 111.6 | 121.0 |
| 2005 | 101.9 | 110.7 | 93.6 | 83.9 |
| 2006 | 104.1 | 86.3 | 106.7 | 105.9 |
| 2007 | 98.7 | 89.4 | 92.7 | 79.5 |
| 2008 | 100.2 | 85.3 | 106.8 | 111.4 |
| 2009 | 100.4 | 111.6 | 100.6 | 100.5 |
| 2010 | 111.5 | 113.4 | 109.0 | 113.9 |
| 2011 | 112.8 | 87.9 | 114.0 | 121.5 |
| 2012 | 86.2 | 110.2 | 89.2 | 69.2 |
| 2013 | 101.4 | 90.3 | 102.6 | 104.7 |
| 2014 | 99.0 | 99.0 | 97.9 | 95.2 |
| 2015 | 96.7 | 99.5 | 99.7 | 97.6 |
| 2016 | 103.3 | 84.4 | 104.9 | 104.1 |
| 2017 | 100.4 | 104.7 | 101.6 | 95.8 |
| 2018 | 108.8 | 105.3 | 112.3 | 94.1 |
| 2019 | 125.3 | 102.3 | 119.0 | 140.6 |
| 2020 | 110.0 | 102.0 | 109.0 | 114.8 |
| 2021 | 108.7 | 104.6 | 111.7 | 114.0 |

We further added the corresponding figure 5 for RpSyu estimates.

[Figure]

**Figure S8: Inter-annual variation of normalized annual recharge (normRpSyu, shown using grey solid dots), precipitation (normP, blue squares), and Ppt-ET (normP-ET, orange squares).**

Reviewer #3 (Other comment 2): Can you comment more in the discussion on the appropriate way to use these data to evaluate models? You make some mention already, but more clear statements on this point would be useful. Such as, temporal variation but not magnitude between these data and recharge estimates, what to do if you have some specific yield numbers to apply for a given area, etc.

Response: We appreciate the reviewer's suggestion to provide clearer guidance on the probable use of the RpSy dataset for model evaluation. In response, we have expanded and modified the discussion to offer more details on how researchers and practitioners can effectively utilize this dataset. We added and modified,

While the RpSy data does not offer direct recharge estimates, it still captures the variations and changes in groundwater recharge over time at daily to coarser temporal resolution. Hence, despite the limitations, uncertainty, and associated caveats discussed in section 2 and 6, the RpSy dataset can be used to validate temporal consistency of recharge estimates derived from empirical methods (Reitz et al., 2017a; Reitz and Sanford, 2019a), physically-based land surface models (Anurag and Ng, 2022; Li et al., 2021; Niraula et al., 2017) or integrated hydrologic models (Kumar and Duffy, 2015; Kollet and Maxwell, 2006; Kumar et al., 2009; Therrien et al., 2010). The RpSy dataset can be utilized for analysing the timing, frequency, and duration of recharge events. Since RpSy provides fluctuations at a daily scale, researchers can use the temporal patterns to assess whether the abovementioned models have the ability to accurately simulate groundwater recharge variability. The match or mismatch between the temporal alignment of RpSy based recharge and model based recharge outputs can provide an assessment of the model's capability to replicate the event based

response to hydroclimatic forcings. Furthermore, the data may also be used to validate the functional relationship between recharge and associated factors as represented in land surface and global hydrologic models. Gnann et al. (2023) demonstrated that theoretical and empirically based functional relationships for recharge differ significantly from global water models. Even when a model produces highly accurate predictions, it may still poorly simulate the strength of functional process couplings. In other words, it may produce right results for the wrong reasons. Such models are likely to underperform during periods when the forcing characteristics are different than those in the training data. The derived benchmark RpSy data, along with forcing variables, can be used to validate the functional relationships represented in models of recharge, using one of the several diagnostic methods such as information theory (Ruddell et al., 2019), causality mapping (Barnett and Seth, 2014; Runge et al., 2019; Runge, 2018), and convergence cross mapping (Ye et al., 2015). The RpSy data may also be used to temporally downscale long-term recharge estimates from observations, thus facilitating generation of recharge inputs for groundwater models (Kim et al., 2008). In circumstances where high confidence specific yield values are available and/or obtainable from field measurements, hydrogeological surveys, or from literature, the RpSy data can be converted into recharge estimates (Recharge = RpSy × Sy). In these cases, a direct comparison can be made between the magnitude of modelled recharge and RpSy based recharge. Additionally, the data may be used for an improved understanding of the role of different forcing and antecedent hydrologic conditions on groundwater recharge and, thus helping manage groundwater aquifers under water-stress conditions. **[Page: 12, Line: 308-333]**

[revised manuscript text omitted]

Reviewer #3 (Other comment 3): Figure 4a and 4c are identical – must be some error in the figure production.

Response: We are grateful to the reviewer for pointing this out. Upon review, it was found that we inadvertently made a mistake during the production of the combined figure, by pasting identical figures in panels 4a and 4c. We have corrected the figure.

[Figure]

**Figure 4: Fraction of recharge in different months and seasons (i.e., Cold seasons (Oct to Mar), Warm-season (Apr to Sept)) relative to the total recharge(/equivalents) for RpSy (top, a and b) and USGS (bottom, c, and d) recharge products. In this plot, USGS recharge data for the grids with RpSy estimates are used. IQR indicates the interquartile range.**

Reviewer #3 The paper has a few minor/grammatical errors and could use a proofreading. A few examples:

L25: strike "the rate of"
L36: "in East Africa is" to "in East Africa are"
L36: "ratio" to "fraction"

Response: We appreciate the reviewer's careful review and attention to detail. We have thoroughly proofread the manuscript and corrected minor and grammatical issues, including the specific changes identified (e.g., removing "the rate of" in L25, changing "in East Africa is" to "in East Africa are," and using "fraction" instead of "ratio" in L36). Thank you for taking the time to point out these errors; your feedback has helped us improve the clarity and overall quality of the manuscript.

---

## Author Response (AR2)

**Legend**

Reviewers' comments

Authors' responses

Direct quotes from the revised manuscript

**Reply to Reviewers' comments (Reviewer#1)**

**Reviewer #1: For final publication, the manuscript should be accepted as is.**

Response: Thank you for your positive feedback and recommendation for final acceptance. We sincerely appreciate the time and effort you invested in reviewing our manuscript. Your thoughtful comments and suggestions helped us clarify our methods and results, ultimately enhancing the overall quality of the work. We are grateful for your support and valuable insights.

**Reply to Reviewers' comments (Reviewer#2)**

Reviewer #2: I thank the authors for their revisions. I think the paper, despite its limitations, makes a useful contribution and should be published. Please consider the below points in this process.

Response: We would like to thank the reviewer for their positive feedback and for recognizing the value of our dataset. We have carefully considered the suggestions made by the reviewer and incorporated them to the best of our ability.

Reviewer #2: the figures still have low quality. Export them in 300dpi and a reasonable format (ideally vector) and they will still come a bit better. This seems like low-hanging fruit to make the work look more professional.

Response: We sincerely appreciate the reviewer's comment. We would like to clarify that all figures were originally produced and exported at a resolution of at least 600 dpi (often higher). However, it appears that the conversion to PDF during the submission process may have inadvertently reduced their quality. In response to the suggestion, we ensured that the figures are submitted at high resolution. We have also submitted the figures separately in high resolution (600 dpi).

Reviewer #2: Figure 1: The fitting of H vs dH/dt is odd. I understand that strong correlations (that are accepted) and weak correlations (that are rejected) exist, but the current fitsshow how well do variations in H fit dH/dt rather than the other way around. As a result, very low correlations get very steep slopes of fitted lines (while normally they should be very horizontal instead). It won't change the overall results, but it's just odd at present.

Response: We thank the reviewer for their note regarding Figure 1. We appreciate their observation that the correlation could be more intuitively represented by interchanging the axes. In the revised

figure, we have swapped the axes to make it consistent with the conventional representation of dH/dt vs. H relation.

Figure 1: (a) Schematic representation of  $\Delta$ H evaluation in the WTF method, which is then used to obtain daily groundwater recharge using Eqn 1, (b) a typical dH/dt vs. H plot used to derive the MRC. The hydrographs (c) and (e) show daily groundwater level (GWL, in m), precipitation (Ppt, in mm), and estimated recharge per unit specific yield (RpSy, in m, discussed later) for a selected and a rejected well, respectively. Here, selection/rejection is based on representativeness of the MRC, which here is determined based on an adj-R2 value of 0.2. (d) and (f) show the dH/dt vs. H plot for the corresponding selected and rejected wells, respectively.

**Reviewer #2: Units of figure 4 are %/month.**

Response: Thank you for your feedback regarding Figure 4. Following the reviewer's concern, we have taken steps to enhance clarity and consistency. Specifically, we have converted the percentage units to fractions and updated the axis labels to "Monthly fraction of USGS recharge" and "Monthly fraction of RpSy."